# Membrane Attack Complex Mediates Retinal Pigment Epithelium Cell Death in Stargardt Macular Degeneration

**DOI:** 10.3390/cells11213462

**Published:** 2022-11-02

**Authors:** Eunice Sze Yin Ng, Nermin Kady, Jane Hu, Arpita Dave, Zhichun Jiang, Jacqueline Pei, Michael B. Gorin, Anna Matynia, Roxana A. Radu

**Affiliations:** 1UCLA Stein Eye Institute and Department of Ophthalmology, David Geffen School of Medicine at UCLA, University of California at Los Angeles, CA 90095, USA; 2Molecular Cellular and Integrative Physiology Interdepartmental Program, University of California, Los Angeles, CA 90095, USA; 3Department of Internal Medicine, Division of Hematology and Oncology, University of Michigan, Ann Arbor, MI 48109, USA; 4Clinical Pathology Department, Faculty of Medicine, Mansoura University, Mansoura 35516, Egypt

**Keywords:** recessive Stargardt disease (STGD1), retinal pigment epithelium (RPE), complement system, bisretinoid-lipofuscin, “disease-in-a-dish”, retinoids, macular degeneration, ABCA4

## Abstract

Recessive Stargardt disease (STGD1) is an inherited retinopathy caused by mutations in the *ABCA4* gene. The ABCA4 protein is a phospholipid-retinoid flippase in the outer segments of photoreceptors and the internal membranes of retinal pigment epithelial (RPE) cells. Here, we show that RPE cells derived via induced pluripotent stem-cell from a molecularly and clinically diagnosed STGD1 patient exhibited reduced ABCA4 protein and diminished activity compared to a normal subject. Consequently, STGD1 RPE cells accumulated intracellular autofluorescence-lipofuscin and displayed increased complement C3 activity. The level of C3 inversely correlated with the level of CD46, an early negative regulator of the complement cascade. Persistent complement dysregulation led to deposition of the membrane attack complex on the surface of RPE cells, decrease in transepithelial resistance, and subsequent cell death. These findings are strong evidence of complement-mediated RPE cell damage in STGD1, in the absence of photoreceptors, caused by reduced CD46 regulatory protein.

## 1. Introduction

Accumulation of toxic bisretinoid-lipofuscin material in the cells of the retinal pigment epithelium (RPE) is an age-dependent process, accelerated by loss or dysfunction of the ATP binding cassette subfamily A member 4 (ABCA4) protein. Mutations in the *ABCA4* gene are responsible for recessive Stargardt disease (STGD1), a juvenile maculopathy that shares many clinical and pathologic features with dry-form of age-related macular degeneration (AMD) [1,2]. Other mutations in *ABCA4* cause cone-rod dystrophy in approximately one-third of cases and serve as rare susceptibility loci for AMD [3,4]. None of these *ABCA4*-mediated blinding diseases are currently treatable. ABCA4 is a flippase for retinaldehyde conjugated to phosphatidylethanolamine (*N*-retinylidene-phosphatidylethanolamine, or *N*-ret-PE) to promote clearance of retinaldehyde and prevent the formation of retinaldehyde dimers (bisretinoids) [5,6]. Recently, the *ABCA4* was shown to be expressed in RPE cells, in addition to photoreceptors [7,8]. ABCA4 in RPE endo-lysosomal membranes is responsible for retinaldehyde recycling during proteolysis of visual pigments following the daily phagocytosis of photoreceptor outer segments (OS). The hypothesized function of ABCA4 in RPE cells was evaluated in vivo using a transgenic mouse line on the *Abca4^−/−^* background that expresses ABCA4 in RPE cells but not in photoreceptors [7]. These transgenic mice exhibited partial rescue of the photoreceptor degeneration and bisretinoid accumulation seen in non-transgenic *Abca4^−/−^* mice, suggesting that ABCA4 in RPE cells reduces the toxicity of retinaldehyde released during proteolysis of rhodopsin following phagocytosis [7]. More recently, we generated RPE cells from molecularly and clinically diagnosed STGD1 patients with altered levels of *ABCA4* transcript and protein as a new “disease-in-a-dish” biological model [9].

Bisretinoids and their oxidative products trigger strong complement reactivity in cultured RPE cells [10]. Further, accumulation of bisretinoids in the RPE of *Abca4^−/−^* mice was shown to activate complement, increase inflammatory markers, and cause oxidative stress [11]. Degeneration of photoreceptors in *Abca4^−/−^* mice and STGD1 patients is thought to result from chronic bisretinoid-mediated complement dysregulation and intracellular buildup of autofluorescent material in the RPE cells [11,12]. Notably, a much greater deposition of the membrane attack complex (MAC) on the basolateral membranes of RPE cells was also found in cadaveric donor eyes from STGD1 patients versus unaffected age-matched human controls [13,14]. In the current study, we used our newly developed “disease-in-a-dish” RPE model from a confirmed STGD1 patient to test the hypothesis that ABCA4 deficiency and complement dysregulation in RPE cells are coupled. We observed a strong correlation between time-dependent deposition of the MAC and RPE cell loss, which mirrored the RPE phenotype detected in *Abca4^−/−^* mice. These findings are the first indication of complement dysregulation and MAC-mediated damage in RPE cells of STGD1 in the absence of the photoreceptor cells. 

## 2. Materials and Methods

### 2.1. Human-Derived RPE Cell Cultures

Intrinsically pluripotent stem cell lines (iPSC) from a clinically and molecularly characterized STGD1 patient (sample “H1” and “H3” in Matynia et al [9]) and an unaffected human donor (control and NDHF sample in Matynia et al. and Lowry et al. studies, respectively [9,15]) were differentiated into RPE cells following published methods [9,16,17,18]. Briefly, undifferentiated iPSCs colonies were passaged using ReLeSR (Stemcell Technologies, Kent, WA, USA) onto a matrigel (1:3 to 1:6, Corning Incorporated, Glenddale, AZ, USA) coated dish with a basal medium containing the DMEM/F12 with Glutamax, NEAA (Millipore Sigma, Burlington, MA, USA), B27, and N2 (Thermo Fisher Scientific, Waltham, MA, USA). The following growth factors were added to the basal medium for the first 48 h: Noggin (50 ng/mL, R & D Systems, Minneapolis, MN, USA), DKK1 (10 ng/mL, R & D Systems) and IGF1 (10 ng/mL, R & D Systems). For the next 48 h, the cells were grown in fresh medium with the above ingredients with the addition of bFGF (5 ng/mL, Thermo Fisher). Days five to six, the cells were grown in the basal medium with DKK1 and IGF1 (10 ng/mL, R & D Systems) and maintained in the basal medium with Activin (100 ng/mL, R & D Systems) and SU5402 (10 μm, Millipore Sigma) for days 7 to 14. At the end of day 14, the cells were switched to Miller medium [19] until the formation of pigmented RPE colonies. The pigmented RPE colonies were picked mechanically and dissociated with 0.25% Trypsin/EDTA (Thermo Fisher Scientific), and then placed onto laminin coated plates until they became confluent in the Miller medium [19]. At passage 2–5, about 2 × 10^5^ total RPE cells (~600,000 cells/cm^2^) were seeded on mouse laminin coated transwell inserts (Millicell-HA) with a pore size of 0.4 μm (Millipore Sigma), allowed to form a monolayer with developed junctional complexes which was assessed by measurement of the transepithelial resistance with an EVOM2 voltohmmeter (World Precision Instruments, Sarasota, FL, USA).

Experimental cultured human RPE cells originated from fibroblasts were collected from a clinically and molecularly characterized STGD1 patient (male) and an unaffected human donor (male). The iPSC lines were validated for viability and karyotype by the UCLA Broad Stem Cell Core using Cell Line Genetics as previously described in Lowry et al [15]. The RPE cells were authenticated using RNASeq to identify the loss of ABCA4 expression in patient cells and presence in Control cells, with key RPE genes comparable between the two cell lines [9]. Control and STGD1 RPE cells were genotyped for single nucleotide polymorphisms (SNPs) associated with AMD and neither of them contained the AMD risk haplotype (Appendix A). 

RPE cells were grown in culture using optimal media *without* retinoids for the initial characterization at two-, three-, and five-months. For longitudinal experiments, media was supplemented with bovine retinal extract, as a source of retinoid (~1.5 pmoles per feeding) for the RPE cells. Supplementation was initiated at two-months under our standard feeding protocol (twice a week) and RPE cells were maintained in culture up to 12-months. Bovine retinal extract was prepared following established protocols [20]. Briefly, retinas were isolated from fresh bovine eyes, sonicated in Ringer’s buffer without calcium and magnesium, agitated overnight at 4 °C, and centrifuged for 20 min at 17,300× *g*. Supernatant was collected and stored at −80 °C. Retinoid content in retinal extract was quantified by normal phase high performance liquid chromatography (HPLC) as described previously [21] and detailed in methods Section 2.8 below.

### 2.2. Animals 

Animals were housed under normal cyclic 12-h light/12-h dark conditions and fed *ad libitum.* Six-months old albino wild-type (BALB/c) and *Abca4^−/−^* on a BALB/c background mice (backcrossed at least eight times) were used for immunostaining and immunoblotting experiments. Equal numbers of male and female per genotype were used for all mouse studies.

### 2.3. Quantitative Real-Time PCR

Total RNA was extracted from RPE cell cultures using Qiagen RNeasy kit (Qiagen Inc., Hilden, Germany) according to the manufacturer’s instructions. NanoDrop (Thermo Fisher) was used to determine total RNA concentration. cDNA was synthesized from one microgram of RNA using the superscript III first-strand synthesis system (Thermo Fisher Scientific). qRT-PCR was performed with SYBR Green (BIO-RAD, Hercules, CA, USA). Primer specificity for stem cell markers (OCT4, NANOG, PAX6, and OTX2), for melanogenesis markers (Tyrosinase, PMEL17 and pigment epithelium derived factor (PEDF)) were used along with housekeeping genes (glyceraldehyde phosphate dehydrogenase (GAPDH), peptidylprolyl isomerase A (PPIA), hydroxymethylbilane synthase (HMBS), and glucose phosphate isomerase 3’ (GPI)). Primer specificity was confirmed with melting temperature analysis and gel electrophoresis. Expression levels were normalized to the geometric mean of all four housekeeping genes: GAPDH, PPIA, HMBS, and GPI. Forward (F) and reverse (R) primer sequences are listed in Appendix A.

### 2.4. In Situ Hybridization Assay

RPE cells were used for in situ hybridization assay with the RNAscope 2.5 HD Chromogenic Detection Kit (Advanced Cell Diagnostics, Hayward, CA, USA) according to the manufacturer’s protocol. Cells were hybridized with human ABCA4 specific oligo probes (Advanced Cell Diagnostics), human RNA polymerase II subunit A probe (Polr2A) as a positive control probe, or bacterial dihydrodipicolinate reductase as a negative control probe, followed by amplification steps and chromogenic detection with Fast Red as previously described [7]. Briefly, Images were captured using a Zeiss Axiophot microscope equipped with a 40× oil-immersion objective lens and a CoolSNAP digital camera (Media Cybernetics, Silver Spring, MD, USA). 

### 2.5. Western Electrophoresis System (WES)

For detection and quantification of ABCA4 protein in RPE cells, capillary electrophoresis was performed on the fully automated Western system (ProteinSimple, Minneapolis, MN, USA) following the manufacturer’s recommendations. Briefly, 20 μg protein lysate were mixed with one μL of the 5× master mix containing sample buffer and DTT. The samples and protein ladder were heat denatured at 95 °C for five minutes. Three μL of the sample were added to the assay plate containing blocking buffer, primary and HRP-conjugated secondary anti-goat or anti-rabbit antibodies (Appendix A), and wash buffer in independent wells, respectively for sequential processing. The plate was briefly centrifuged and loaded into the instrument for electrophoretic separation of proteins in capillary tubes containing a 66- to 440-kDa separation matrix. Protein levels were quantified using the automated Compass software according to the manufacturer’s guidelines (ProteinSimple). 

### 2.6. Western Blot (WB)

Mouse RPE along with Bruch’s membrane and choroid were harvested from six-month-old albino wild-type and *Abca4^−/−^* mice, both on a BALB/c background. Human cultured RPE were grown on Millicell-HA for three months. Both mouse and cultured human RPE cells were homogenized in 1× PBS with Halt protease inhibitor mixture (Thermo Fisher Scientific). Protein samples were treated with benzonase nuclease (Millipore-Sigma) at room temperature (RT) for one hour and re-homogenized with 1% SDS. Cellular debris was removed by brief centrifugation, and protein concentration was determined using the Micro BCA Protein Assay Kit (Thermo Fisher Scientific). 25 μg of total protein of mouse homogenate or three μg of total protein of RPE homogenate or 40 μg cell concentrated media, respectively was then separated on 4–12% or 12% SDS/PAGE gels (Novex; Thermo Fisher Scientific). PVDF membranes were blocked with Odyssey Blocking Buffer (LI-COR Biosciences, Lincoln, NE, USA), followed by incubation at RT, probed with primary antibodies and cognate IR dye-labeled secondary antibodies (Appendix A) [6,22,23,24]. Western blot analysis was done using the Odyssey CLx Infrared Imaging System (LI-COR). Colored images were converted to grayscale for viewing using Photoshop (Adobe, San Jose, CA, USA). 

### 2.7. Exogenous Retinoid Uptake and Processing by RPE Cells

RPE cells with TER of >150 Ω/cm^2^ grown in media *without* supplementation of bovine retinal extract for ~three-months were used in functional assays (visual cycle enzymatic steps and ABCA4-flippase activity). All experiments involving retinoids were conducted in a dark room under dim red light. To assay visual cycle activities, RPE cells were incubated overnight at 37 °C with 10 µM all-*trans*-retinol (MilliporeSigma) in DMEM (Thermo Fisher Scientific) supplemented with 1% BSA (bovine serum albumin, fatty acid–free) on the basal side. Apo inter-photoreceptor retinoid-binding protein (IRBP, 10 μM) was added to the apical medium to protect the released 11-*cis*-retinoids. IRBP was purified from freshly dissected bovine retinas and apo-IRBP was obtained after retinoids were degraded by exposure to UV-light [20]. To evaluate the ABCA4-flippase activity, we incubated overnight the RPE cells with bovine OS applied to the apical side (~75 × 10^6^ OS per transwell; ~1.2 nmoles total retinaldehyde). Cultured cells on filters were washed with 1× PBS buffer and stored dry at −80 °C until further processing. Corresponding media samples were collected in tubes containing hydroxylamine (final concentration 200 mM) and processed immediately for retinoid extraction.

### 2.8. Retinoid Extraction and Normal-Phase Liquid Chromatography Analysis

RPE cells were homogenized in phosphate buffer containing 200 mM hydroxylamine. Cell homogenates and media samples containing hydroxylamine were mixed by vortexing and incubated at RT for ~20 min. Two mL of methanol was added, and retinoids were extracted twice with three mL of hexane followed by centrifugation at 3000 × *g* for five minutes. The organic phases were collected, dried under a stream of argon gas, and re-dissolved in 100 μL of hexane. Hexane solutions were analyzed by normal phase HPLC as previously described [21]. The identity of each retinoid-eluted peak was established by comparing the spectra and elution times with those of authentic retinoid standards. Retinoids standards were obtained from Sigma, and retinaldehyde oximes were rederived in house by hydroxylamine reduction and purified by HPLC [25]. Peak area for retinol and retinyl esters were acquired at 325 nm and retinaldehyde oximes at 350 nm, according to their maximum absorbance spectra. Sample peaks were quantitated by comparing peak areas to calibration curves established with retinoid standards using the published molar extinction coefficients [25,26]. 

### 2.9. Phagocytosis Assay

Pulse-chase phagocytosis assay was done with RPE cells grown as monolayers on culture inserts incubated with bovine photoreceptor OS (Invision BioResources, Seattle, WA, USA) (10 OS/cell; total ~2.5 × 10^6^ OS per transwell) in the apical medium for two hours. After the initial OS challenge, cells were washed with Dulbecco’s Phosphate Buffered Saline (DPBS), containing 0.9 mM Calcium Chloride and 0.49 mM Magnesium Chloride (DPBS-CM), and immediately processed for immunofluorescence (pulse), or incubated further for two hours and then processed for immunofluorescence (chase) as previously described [27]. A double immunofluorescence labeling strategy, using a primary antibody against rhodopsin (Rho), to distinguish between OS bound to the surface versus OS that has been internalized by the RPE cells. Briefly, cultures were fixed with 4% formaldehyde for ten minutes at RT and blocked with 1% BSA in DPBS-CM for 15 min. Surface-bound OS was labeled with the mouse anti-Rho 1D4 [27] conjugated with Alexa Fluor 488-nm goat anti-mouse secondary antibody (Appendix A). After permeabilization with 50% ethanol in DPBS-CM for five minutes, OS was labeled with the same Rho 1D4 antibody conjugated with the Alexa Fluor 568-nm goat anti-mouse secondary antibody (Appendix A). Finally, cells were washed with DPBS-CM and the membranes of the culture inserts were excised and mounted onto microscopy slides. Confocal z-stacks of randomly selected fields were captured using an Olympus FluoView FV1000 confocal microscope with a 60× NA1.4 oil objective. Surface-bound OS particles were labeled with both secondary antibodies (green and red), thereby appearing yellow. Internalized OS particles were labeled only with the Alexa Fluor 568-nm-conjugated secondary antibody (red only). For quantification, OS particle with a ≥0.5 μm diameter were counted from a total of five to ten fields of view using NIH Image J software (Rasband, W.S., ImageJ, U. S. National Institutes of Health, Bethesda, Maryland, USA; Available online: https://imagej.nih.gov/ij/, accessed on 4 March 2019). Analysis of OS degradation (ingested) was estimated after subtracting the OS bound-only from the total number of OS particles after permeabilization of the cells.

### 2.10. Mouse Immunohistochemistry

Age-matched six-months albino wild-type and *Abca4^−/−^* were euthanized and the superior part of the eyeballs were marked with a cautery pen. The eyes were enucleated and fixed in 4% formaldehyde with 0.1 M sodium phosphate buffer (NaPO_4_, pH 7.4) for 30 min at room temperature (RT) for OD eye and overnight for OS eye. After washing, the eyes were dissected in 0.1 M NaPO_4_ and the anterior segment (cornea, lens, vitreous) and the neural retina were removed to obtain the RPE-choroid-scleral eyecups. The OD eyecup was flattened by cutting into eight leaflets to obtain an RPE-choroid-scleral flatmount. The OS eyecups were infiltrated with 10–30% sucrose for cryoprotection, embedded in Optimal Cutting Temperature embedding medium (OCT; Tissue-Tek, Torrance, CA, USA), and cut into 10-µm frozen sections. The RPE flatmounts and frozen sections were permeabilized with 1% Triton X-100 at RT for 1 hour and 10 min, respectively, then blocked with 1% BSA/5% goat serum/0.5% TritonX-100 at 4 °C overnight for flatmounts and one hour at RT for frozen sections. Primary antibodies were incubated at 4 °C for two days and overnight for RPE flatmounts and frozen sections, respectively. Following three washes with PBS/ 0.1% TritonX-100, secondary antibodies were applied at RT for two and one hour to flatmounts and frozen sections, respectively. The RPE cells were delineated by the phalloidin conjugated with Texas red, staining the actin filaments, with nuclei stained by DAPI (300 nM, Thermo Fisher Scientific), and then mounted with 5% n-propyl gallate in 100% glycerol. Images from superior temporal region, at similar distance from the optic nerve, of the RPE flatmounts and frozen sections were captured with an Olympus FluoView FV1000 confocal microscope using a 60× oil-immersion objective. The z-orthogonal sections images were obtained using Imaris software (Bitplane Inc., Concord, MA, USA). 

### 2.11. RPE Cell Immunocytochemistry (ICC)

Cultured RPE cells with their supporting filter were fixed in 4% formaldehyde and 0.1 M PBS for 30 min at RT. For confocal microscopy analysis of RPE monolayer cross-sections, the fixed cells along with their filter were embedded in agarose (Millipore-Sigma) and 50 μm sections were cut with a VT1000s vibratome (Leica Microsystem, Wetzlar, Germany) [28]. To visualize phosphatidylethanolamine in the cell membranes, RPE cells were incubated for two hours at RT with 1 µM Duramycin conjugated to Biotin (Molecular Target Technologies, West Chester, PA, USA) diluted from 200 µM stock solution in water containing 5% DMSO. For both wholemount and agarose sections of cultured RPE cells, primary antibodies were applied overnight at 4 °C followed by the secondary antibodies for 1 hour at RT using either goat anti-mouse IgG or goat anti-rabbit IgG, or streptavidin conjugated to Alexa Fluor dyes (Thermo Fisher Scientific). In addition, phalloidin and DAPI were used to delineate the cell boarder and nuclei, respectively. The cells were mounted with 5% n-propyl gallate in 100% glycerol with coverslips. All primary and secondary antibodies source and concentration were listed in Appendix A. Images were captured with an Olympus FluoView FV1000 confocal microscope using a 60× oil-immersion objective. Images were visualized and analyzed by Imaris (Bitplane Inc.) or FluoView software (Olympus, Waltham, MA, USA). 

### 2.12. Transmission Electron Microscopy

RPE cells grown on culture inserts (Millicell-HA) were fixed with 2% formaldehyde and 2.5% glutaraldehyde in 0.1 M PBS (pH 7.2), treated with 1% osmium tetroxide dissolved with 0.1 M PBS, then dehydrated in a graded series of alcohols and embedded in Araldite 502 (Electron Microscope Sciences, Hatfield, PA, USA). Ultrathin sections (at 70 nm in thickness) were cut on a Leica Ultracut microtome, collected on 200-mesh copper grids, and double-stained with uranyl acetate and lead citrate. The images (~12–18 per transwell) were collected in a JEM-1400 electron microscope (JEOL, Peabody, MA, USA) with ORIUS SC1000b Camera at 2000× and analyzed using Gatan Microscopy Suite Software (Gatan, Inc., Pleasanton, CA, USA).

### 2.13. Quantification and Statistical Analysis

Statistical analysis for each experiment is described in figure legends, where “n” represents biological replicates. For human studies, we used one line of Control and two lines of STGD1 RPE cells seeded and grown on filters in 24-transwell plates; one filter (with ~2 × 10^5^ RPE cells per filter/transwell) represents an individual biological sample, unless otherwise specified. Every experiment with human RPE cells included a minimum of three independent biological samples and the experiments were repeated twice with triplicates (3 × single filter/transwell for a total of n = 6), or with duplicates (2 × single filter/transwell, for a total of n = 4) per group/genotype. The results were presented as means with standard deviation of a minimum of three to six animals or human biological samples per group/genotype. Data analyses were performed using GraphPad Prism 9.0 package (GraphPad by Dotmatics, San Diego, CA, USA). Comparisons between groups were evaluated using two-tailed Student’s *t*-test with a statistical significance reported at * *p* < 0.05; ** *p* < 0.005; *** *p* < 0.0005; **** *p* < 0.0001.

## 3. Results

### 3.1. Control and STGD1 RPE Cultured Cells Have Similar Morphological Features

In vivo complement activation observed in *Abca4^−/−^* mice and excessive MAC deposition evidenced in STGD1 postmortem donor eyes compelled us to investigate the role of complement in STGD1 pathogenesis using patient-derived RPE cells [11,12,13]. For this study, we selected a STGD1 patient with two ABCA4 mutations on different alleles: (*i*) c.3386G > T; p.Arg1129Leu and (*ii*) c.[5461-10T > C;5603A > T]; p.[Thr1821Aspfs*6,Thr1821Valfs*13;(Asn1868Ile)] [29,30,31]. The R1129L substitution on the first allele is located within the nucleotide-binding domain 1 (NBD1) of ABCA4, previously shown to cause reduction in protein abundance and ATP-binding capacity when expressed in the 293T cells [32]. The second allele mutation was newly identified in this STGD1 patient and predicts a null protein [29]. The iPSC lines of control (normal) and clinically diagnosed STGD1 patient were derived into RPE cells according to standardized protocols [9,16,27] with their molecular characterization being recently published in Matynia et al [9]. In the current study, the RPE cells were seeded on a transwell filter and maintained in culture using a conditioned medium *without* retinoids for an initial morphological evaluation. By qRT-PCR analysis, mRNA for differentiation and pigmentation marker proteins were at similar levels in control and STGD1 RPE cells at three- and five-months in culture, while the mRNAs for pluripotency markers were insignificantly expressed (Appendix A). By a chromogenic in situ hybridization assay, we observed a comparable distribution of ABCA4 mRNA expression in the STGD1 and control RPE cells (Appendix A). Similar levels of ABCA4 mRNA in STGD1 and control RPE cells were found by qRT-PCR after two- and five-months in culture (Figure 1A), consistent with near-normal ABCA4 transcript levels found by RNASeq analysis [9]. At the protein level however, STGD1 RPE cells contained only ~15% of the ABCA4 found in the control RPE cells (Figure 1B and Appendix A). By immunohistochemistry, control RPE cells displayed a homogeneous membrane distribution of ABCA4 throughout, unlike the punctate profile observed in the STGD1 RPE cells (Figure 1C). Intracellular localization of normal ABCA4 protein in RPE cells was assessed with an antibody against the endosome marker EEA1 (Appendix A), as previously shown in mouse RPE cells and fetal human RPE cultured cells [7]. In STGD1 RPE cells, the mutated ABCA4 protein seemed to colocalize with EEA1 (Appendix A). Together, these findings suggest degradation and mislocalization of the mutated ABCA4 protein. At two-months in culture, control and STGD1 RPE cells had both grown to confluent monolayers with homogenous pigmentation and a typical RPE cobblestone appearance (Figure 1D). Transmission electron microscopy revealed polarized RPE cells with apical microvilli, basal infoldings, and intracellular pigment granules for both STGD1 and control RPE cells (Figure 1E and Appendix A). En-face confocal images of phalloidin staining for apically localized actin filaments showed well-defined hexagonal cell borders with similar cell counts for control and STGD1 RPE cells (Figure 1F,G). Since the formation of functional cellular junctions is crucial for both maintenance of epithelial integrity and the barrier role of RPE, we measured the transepithelial resistances (TER) in three-months cultures. STGD1 and control RPE cells exhibited similar TER of ~200 Ω/cm^2^ (Figure 1H), comparable to the TER of human fetal RPE cells in culture [27,33,34]. 

### 3.2. OS Binding, Internalization, and Rhodopsin Degradation Are Normal in STGD1 RPE Cells

Phagocytosis of distal photoreceptor OS is an important function of RPE cells, which serves to remove toxic products and maintain photoreceptor health [35,36]. To determine if RPE cells are capable of phagocytosis, two-months cultured STGD1 and control RPE cells were incubated with purified bovine OS using an optimized pulse-chase assay [27], in which the amount of surface-bound Rho-containing OS (green channel) was determined before permeabilization of the cells, and the amount of ingested Rho-containing OS was determined after permeabilization of the cells as the difference between total Rho-containing OS (permeabilized cells, red channel) and bound Rho-containing OS (green channel). In the pulse phase (2 h, Appendix A), staining with an antibody against rhodopsin (Rho) before (Appendix A, Rho-bound) and after permeabilizing the cells (Appendix A, Rho-total), we observed similar amounts of both bound (green arrows in all images) and ingested Rho-containing OS (white arrows in the red channel and merge images) in the STGD1 and control RPE cells (Appendix A). In the chase phase (2 h after removal of bovine OS from the media), we observed degradation of internalized Rho and an overall decrease in Rho staining for both control and STGD1 RPE cells (Appendix A). Quantification of total Rho showed no significant difference between control and STGD1 RPE cells, suggesting similar rates of OS uptake and Rho proteolysis (Appendix A). 

### 3.3. ABCA4-Flippase Activity Is Reduced in STGD1 RPE Cells

To test for ABCA4-flippase activity, RPE cells grown in culture for three-months *without* exposure to retinoids were fed with a single dose of bovine OS, estimated to contain ~1200 pmoles 11-*cis*-retinaldehyde (11c-RAL) in the form of rhodopsin. Media and cells were harvested the following day (Figure 2A). Upon proteolytic digestion of OS proteins in RPE phagolysosomes, free 11c-RAL is released and thermally isomerized to all-*trans*-retinaldehyde (at-RAL) within acidic endolysosomes. ABCA4 was shown to ‘flip’ 11c- and at-RAL coupled to phosphatidylethanolamine (as N-ret-PE) across membranes with similar efficiencies [5]. The transfer of N-ret-PE from the lumenal to cytoplasmic face of the membranes, in photoreceptor OS and RPE endolysosomes, prevents the secondary condensation of N-ret-PE with another at-RAL to form a toxic bisretinoid. In RPE cells, translocation of N-ret-PE facilitates the recycling of at-RAL to 11c-RAL visual chromophore via the RPE visual cycle [5,7] (Figure 2B). After overnight incubation, we extracted retinoids from both the media and RPE cell-homogenates and analyzed them by HPLC. Untreated cells and their corresponding media showed *no* detectable retinoids (Figure 2C and Appendix A). The retinoid content of OS-supplemented media was similar for control- and STGD1-derived RPE cell cultures, mainly comprising 11c-RAL (Appendix A). In contrast, HPLC analysis of the RPE cell homogenates showed approximately two-fold lower levels of all-*trans*-retinyl-palmitate (at-RP), an insoluble storage form of vitamin A, in STGD1 vs control RPE cells. This result is consistent with the reduced N-Ret-PE flippase activity of the mutated ABCA4 protein and impaired recycling of OS visual pigments (Figure 2C,D). PE phospholipid alone is also a substrate of ABCA4 that was shown to be actively flipped across the photoreceptor disc membranes in the same direction as N-Ret-PE [5]. Media of two-month-old cultured RPE cells was supplemented with bovine retina extract and the PE distribution was visualized by confocal microscopy at six-months after a short incubation with duramycin, a cyclic peptide that binds PE with high affinity and specificity [37]. We found that duramycin-bound PE was dispersed along both plasma and internal membranes of the control RPE cells (Figure 2E). In contrast, STGD1 RPE cells displayed intracellular PE-aggregates without clear PE-duramycin-association with the plasma membrane (Figure 2E and Appendix A). Quantification of pixel intensity from confocal images stained with PE-duramycin indicated ~50% increased levels in the RPE cells of STGD1 versus control (Figure 2F), which is further evidence of dysfunctional ABCA4 protein.

### 3.4. LRAT and RPE65 Activities Are Similar in Control and STGD1 RPE Cells

Reduced levels of at-RP in homogenates of STGD1 versus control RPE cells (Figure 2D) could be caused by reduced LRAT protein levels instead of reduced at-ROL substrate for at-RP synthesis. To test this possibility, we performed quantitative immunoblotting and found similar LRAT levels in the control and STGD1 cell homogenates (Appendix A). Immunoreactivity for LRAT antibody in agarose sections showed comparable staining in both STGD1 and control RPE cells (Appendix A). Still another possibility is that RPE65 level and activity may be increased in STGD1 versus control cells. Since RPE65 is a retinoid isomerohydrolase that converts at-RP to 11c-ROL plus free palmitic acid [38], increased RPE65 could also cause reduced levels of at-RP in STGD1 RPE cells. We observed similar levels of the RPE65 protein in control and STGD1 RPE cells as determined by quantitative immunoblotting and confocal microscopy, respectively (Appendix A). To evaluate the specific activities of LRAT and RPE65 in vivo, we incubated the RPE cells with 10 μM at-ROL in the basal media supplemented with 1% BSA (Appendix A). Additionally, the apical medium was supplemented with 10 μM of apo-interphotoreceptor retinol-binding protein (apo-IRBP) to protect the released 11c-RAL generated after at-ROL was processed through the RPE visual cycle steps. IRBP has high affinity for 11c-RAL facilitating its transfer from the RPE to photoreceptors to regenerate the visual pigment by binding to the cone- and rod-opsins [39,40]. Extracts of the RPE cell homogenate and corresponding media were analyzed by HPLC to quantify the retinoid content. We found that levels of at-ROL and at-RP, the resultant product of LRAT activity, were similar in the homogenates of control and STGD1 RPE cells (Appendix A). Also, no difference in the levels of at-ROL and 11c-RAL released in the media was observed in the control vs STGD1 RPE cells (Appendix A). Taken together, these data suggest that the levels and activities of the key visual-cycle proteins, LRAT and RPE65, were similar in control versus STGD1 RPE cells supplied with at-ROL, and therefore not responsible for the reduced at-RP found in the STGD1 cells fed with OS containing retinaldehydes (Figure 2D). 

### 3.5. Autofluorescent Material Accumulates in STGD1 RPE Cells

Impaired recycling of retinaldehyde, due to loss of the ABCA4 transporter function, leads to secondary condensation of *N*-ret-PE with at-RAL to form an autofluorescent bisretinoid such as A2E in the RPE cells of STGD1 donors and *Abca4^−/−^* mice [13,41]. Here, we acquired bisretinoid-autofluorescence images at 488nm of control and STGD1 RPE cells maintained in culture for three- or 12-months in the presence of bovine retinal extract (~1.5 pmoles RALs per feeding twice per week). Compared to RPE cells from unaffected human controls, STGD1 RPE cells showed slightly increased autofluorescence at three-months, and two-fold increased autofluorescence after 12-months in culture (Figure 3A,B). In *Abca4^−/−^* and STGD1 donor eyes, the presence of bisretinoids in the RPE caused cellular stress, evidenced by elevated lipid peroxidation products such as malondialdehyde (MDA) and 4-hydroxynonenal (4-HNE) [11,13]. Similarly, analysis of STGD1 RPE cells grown in culture for 12-months showed stronger immunoreactivity for the 4-HNE antibody compared to the control RPE cells (Figure 3C). 

### 3.6. Complement Dysregulation Is Evidenced in STGD1 RPE Cells

Stimulation of C3 activity by bisretinoids has been shown in vivo and in vitro [10,11,42]. Here, we assessed C3 components and the complement negative regulatory proteins, CD46, CD59, and CFH by immunohistochemistry and immunoblotting at three-months in human-derived RPE cultured cells. On sections, C3 immunolabeling appeared throughout the RPE cells and was approximately two-fold higher in STGD1 versus control RPE cells (Figure 4A,B). Elevated C3b/iC3b breakdown fragments were further confirmed by quantitative immunoblot analysis (Figure 4C,D). Using fixed sections and RPE cell homogenates, C3a, a short-lived soluble fragment released after C3 fragmentation, was investigated with an antibody against its more stable form, C3a-desArg along with its C3a cognate membrane-bound receptor C3aR. While the C3a fragment was not detected in the RPE cells (Figure 4C), immunoblotting and immunostaining for its cognate receptor, C3aR, showed similar distribution and levels in control and STGD1 RPE cells (Appendix A). C5aR levels were also similar in control and STGD1 RPE cells (Appendix A), suggesting that the C5a/C5aR signaling pathway downstream of C3a/C3aR is also not activated. Next, we determined the levels of CD46, a membrane-bound, early-stage inhibitor of C3 convertase. CD46 appeared to localize predominantly on the basolateral side of the RPE cells. By confocal microscopy, CD46 immunostaining was notably diminished and pixel intensity quantification of CD46 showed approximately two-fold reduction in STGD1 versus human control RPE cells (Figure 4E,F). Further, CD46 levels in STGD1 RPE cells were only ~65% of the levels in control RPE cells by quantitative immunoblotting (Figure 4G,H). Unlike CD46, the cellular distribution and levels of CD59, another membrane-bound complement regulatory protein that inhibits the final step of the C3 cascade, were similar in the STGD1 and control RPE cells (Appendix A). By quantitative immunoblotting, we measured levels of complement factor H (CFH), a major soluble complement regulatory protein, in the media of cultured RPE cells. Levels of CFH, along with unrelated secreted protein PEDF, were also similar between STGD1 and control RPE cells (Appendix A). Taken together, these data suggest a primary deficiency of C3 convertase inhibition during the early complement amplification phase on the surface of STGD1 RPE cells.

### 3.7. Markedly Elevated MAC Deposition Is Present on the STGD1 RPE Cells

Excessive C3 hydrolysis and increased C3b opsonizing affinity on the plasma membrane due to C3 convertase activity are responsible for the assembly of terminal complement complex C5-C9 [43]. We assessed formation of this terminal complex on the surface of STGD1 and control RPE cells at different times in culture by confocal microscopy with an antibody against MAC (C5b-9 complex). We observed 1.2- and two-fold increased MAC levels on the STGD1 versus control RPE cells at three and 12 months in culture, respectively (Figure 5A,B and Appendix A). Orthogonal images obtained from z-stacked optical sections, from the apical side, showed that MAC deposition was predominantly on the basolateral poles of these cells and evidenced its internalization more abundantly in the STGD1 RPE cells (Figure 5A). Additionally, MAC localization was confirmed in the STGD1 and control RPE cells using an antibody against peropsin, an integral membrane protein expressed *only* in the apical microvilli of RPE cells (Appendix A) [22]. Furthermore, the en-face view of phalloidin-staining apically distributed actin filaments showed an advanced dysmorphism (discontinued lines and wider cell size, indicated by stars, respectively) in the aged STGD1 versus control RPE cells (Figure 5A and Appendix A). These changes after 12-months in culture were mirrored by a decline in junctional complexes between cells, reflected by a ~35% loss of cell number and ~80% reduction of TER in the STGD1 versus control RPE cultures (Figure 5C,D).

### 3.8. MAC-Mediated RPE Cell Damage Is Evident in the Eyes of Abca4^−/−^ Mice

Accumulation of toxic bisretinoids and elevated complement C3 activity were previously shown in the RPE of *Abca4^−/−^* mice at ~10-weeks, prior to the onset of photoreceptor degeneration [11,41]. Presumably, oxidized bisretinoids and C3b opsonization on the plasma membrane, along with the buildup of internalized C3b/iC3b fragments, led to a loss of RPE cells followed by degeneration of photoreceptors beginning at six-months in *Abca4^−/−^* mice [41,44,45]. Here, we evaluated the C3b/C3a breakdown fragments and their corresponding inactive forms (iC3b/C3a-desArg), by immunoblotting of RPE homogenates from six-month-old wild-type versus *Abca4^−/−^* mice (Appendix A). We performed this analysis using a C3 antibody against the full-length protein and a C3a-desArg antibody specific for the smaller fragment C3a. Irrespective of the antibody and using reducing conditions to render all C3 fragments in the RPE cell homogenates, we could not detect the C3a ~9 kDa band in the mouse RPE homogenates from either wild-type or *Abca4^−/−^* mice (Appendix A). Moreover, immunostaining of mouse retina with C3a-desArg antibody displayed comparable C3a intensity for both wild-type and *Abca4^−/−^* mice (Appendix A). At the same time, levels of the C3a receptor protein (C3aR) were similar in wild-type and *Abca4^−/−^* RPE homogenates by immunoblotting (Appendix A), and in mouse retina sections by confocal immunofluorescence analysis (Appendix A). To investigate the extent of the complement activation, we prepared RPE flatmounts from wild-type and *Abca4^−/−^* mouse RPE and performed immunohistochemistry with an antibody against C5b-9 terminal complement complex. We observed more intense immunolabeling of C5b-9 complex in RPE cells from *Abca4^−/−^* mice versus wild-type mice (Appendix A). Further, we detected larger cells containing three or more nuclei, likely due to membrane fusion between cells, suggesting the formation of RPE syncytia in *Abca4^−/−^* compared to wild-type mouse flatmounts (Appendix A). Orthogonal z-stacked optical sections, obtained from the apical side of the RPE flatmounts, revealed that MAC deposits were internalized by the RPE and were more pronounced on the basolateral plasma membrane of the *Abca4^−/−^* mice (Appendix A). Cell count showed a two-fold increase in the multinucleated cells in the *Abca4^−/−^* vs wild-type RPE flatmount (Appendix A). These data suggest that in the *Abca4^−/−^* mice, the C3 amplification loop is accompanied by an ongoing fully activated complement cascade and possible MAC-mediated lysis of RPE cellular plasma membrane. 

## 4. Discussion

The RPE is an integral component of the blood-retina barrier and contributes to daily maintenance of the adjacent cellular layers [46]. The initiator of photoreceptor degeneration in STGD1, and in some cases of AMD that are associated with specific ABCA4 variants, is thought to be due to RPE dysfunction [3]. Rod and cone photoreceptors daily shed their distal OS, which are phagocytosed by the apical RPE as a mechanism to recycle OS components and to detoxify lipids that have undergone oxidation and other light-dependent modifications [47,48]. Following phagocytosis, rhodopsin and cone-opsin visual pigments undergo proteolysis, liberating their chromophore as free retinaldehyde, which is cytotoxic and highly reactive [49]. Until recently, little was known about the detoxification and recycling of retinaldehyde released into RPE phagolysosomes. When ABCA4 is missing, in *Abca4^−/−^* mice or STGD1 patients, fluorescent retinaldehyde adducts (bisretinoids including A2E) accumulate within RPE cells [41,50]. Recently, ABCA4 was shown to be present in RPE internal membranes, in addition to photoreceptor OS [7,8]. Together these findings suggest that ABCA4 in RPE serves to recycle retinaldehyde released during degradation of visual pigments by transferring *N*-ret-PE from the luminal to cytoplasmic face of endolysosomal membranes where the retinaldehyde can be reduced to retinol and re-enter the RPE visual cycle. 

In the current study, we analyzed the role of ABCA4 in RPE cells from a clinically and genetically diagnosed STGD1 patient and an unaffected human control. Characterization of RPE cells from these individuals showed that the ABCA4 products of the mutant loci had no effect on the growth of fibroblasts, dedifferentiation of fibroblasts into iPSCs, or reprogramming of iPSCs into RPE cells (Figure 1 and Appendix A, and [9]). Importantly, both control and STGD1 RPE cells matured into confluent monolayers at the same rate with similar trans-epithelium resistances and displayed normal ultrastructural features. Further, neither the control nor STGD1 RPE cells contained the AMD-associated SNP haplotypes (Appendix A). Lastly, both the OS phagocytotic capacities and visual-cycle enzyme activities were similar in control and STGD1 RPE cells, allowing us to interrogate the coupling of ABCA4 deficiency and complement dysregulation as the driver of RPE cell death without the input from photoreceptor cells.

Although ABCA4 mRNA levels were comparable for STGD1 and control cells, levels of the ABCA4 protein were five-fold lower in the STGD1 cells signifying functional consequences [9,29,32]. We tested the ABCA4 N-Ret-PE flippase activity in the control and STGD1 RPE cells that were *never* previously exposed to visual retinoids, by feeding them with a single ‘meal’ of bovine OS. As predicted by the patient’s ABCA4 variants, recycling of retinaldehyde was significantly reduced in the STGD1 RPE cells. ABCA4 also functions as a flippase for PE unconjugated to retinaldehyde [5], signaling a potential role in the cellular distribution of PE. To test this possibility, we investigated PE localization in RPE cells following treatment with duramycin, which binds the headgroup of PE [37]. Intracellular PE-aggregates and reduced PE in the plasma membrane in STGD1 RPE cells suggest that altered PE metabolism may play a role in STGD1 pathogenesis. This novel finding demands further investigation using additional RPE cells obtained from STGD1 patients with different ABCA4 mutations. 

Patients with *ABCA4*-mediated retinopathies exhibit a strong correlation between visual loss and increased short wavelength (SW) autofluorescence in the RPE due to the presence of bisretinoids [2]. Bisretinoid levels are also several-fold higher in the RPE of *Abca4^−/−^* versus wild-type mice [7]. Although ABCA4 is far more abundant in photoreceptors versus RPE cells, bisretinoid levels are several-fold higher in the RPE versus neural retina [7]. Further, the major bisretinoid species, A2E, is undetectable in *Abca4^−/−^* neural retinas and only found in RPE cells. Here, RPE cells supplemented with bovine retinal extract showed much more 488-nm autofluorescent material in STGD1 versus control cells. *In vivo*, the bisretinoid chemical structure renders them susceptible to further modification and fragmentation, thereby generating reactive species responsible for increased oxidative stress and destabilization of cellular physiological activities [51,52,53,54,55,56,57]. Similar to STGD1 donor eyes [13], 12-month-old STGD1 RPE cells showed markedly increased levels of the lipid peroxidation marker, 4-HNE. Thus, RPE cells from the STGD1 patient recapitulates multiple features of the phenotypes seen in STGD1-human and *Abca4^−/−^* mouse eyes. 

The complement system is a proteolytic cascade, activated through several initiating pathways, representing the first line of defense against an invading pathogen [43]. To prevent inappropriate inactivation and an attack on self, C3 activation, C3 hydrolysis (tick-over), and C3 degradation are all kept in check by multiple complement negative-regulatory proteins including CD46, CD59, and CFH. Both activating and regulatory proteins of the complement cascade were shown to be expressed in RPE cells, suggesting that the RPE protects the eye from blood-borne pathogens without dependence on systemic factors [58,59,60]. Disruption of RPE homeostasis can lead to C3-mediated attack on these cells. Augmented autofluorescence build-up observed in STGD1 RPE cells triggered C3 activity in a similar fashion as was shown in the RPE cells of *Abca4^−/−^* mice and cultured human fetal RPE cells challenged with bisretinoids [11,61,62]. In a previous study, we showed that over-expressing the complement receptor 1-like protein y (CRRY), a major murine complement regulator and a functional homolog of human CD46, reduced complement attack on the RPE and rescued both bisretinoid accumulation and photoreceptor degeneration in *Abca4^−/−^* mice [12]. Here, increased C3 convertase activity and reduced levels of the CD46 negative-regulatory protein set up the RPE for attack by the complement system in STGD1 cells while CFH levels remain unchanged. These findings suggest that photoreceptor degeneration in *Abca4^−/−^* mice and STGD1 patients results from loss of RPE support due to chronic complement dysregulation in the early stage of the cascade. 

The RPE’s ability to internalize C3 and C5b-9 complex is another built-in defensive mechanism to prevent further C3b opsonizing fragments and formation of a cytolytic MAC on the plasma membrane [11,12,63]. Despite similar cell densities and morphologies, we observed substantial MAC deposition on the surface of STGD1 versus control RPE cells at three-months. MAC lytic effect is potentiated by the insertion of additional C9 molecules into the C5b-9 terminal complex blocked by CD59, another membrane-bound negative regulator [64]. Here, we found that CD59 expression was not altered in STGD1 RPE cells signifying further that the early inhibitory step of the complement cascade by CD46 is essential. When cells were maintained in culture for ~one year, we observed extensive MAC-mediated damage to membranes with ~35% loss of cells in the STGD1 RPE cultures. Finally, the barrier integrity of RPE cells derived from the STGD1 patient was severely compromised, indicated by a several-fold drop in TER. Comparable MAC-mediated pathological features were recently described in three donor eyes from STGD1 patients [13,14]. Notably, an inverse correlation between the basal laminar autofluorescent deposits in dysmorphic RPE cells and CD46 immunoreactivity was also evidenced in the AMD donor eyes [65]. STGD1 and AMD, despite superficially unrelated causes (loss of the ABCA4 transporter versus multigenic risk alleles including mutations in the genes for complement regulatory proteins), may actually both be caused by chronic inflammation of the RPE due to dysregulation of the complement system. 

Taken together, these findings stress the importance of the CD46 regulator to limit the downstream C3 convertase activity that can lead to the harmful levels of complement terminal complex on the RPE cells. In STGD1, the ongoing intracellular PE-buildup and autofluorescent bisretinoid material, due to dysfunctional ABCA4 flippase, promotes the complement activity and drives the formation of the terminal complement complex C5b-9 (Figure 6). It appears that chronic deposition of the C5b-9 terminal complex, initially at a sub-lytic level, eventually destabilizes the plasma membrane by forming a lytic pore with loss of the RPE cells. Inefficient PE translocation by mutated ABCA4 may also compromise RPE cellular integrity due to an abnormal phospholipid composition of the plasma membrane. In STGD1 and sub-group of AMD patients, loss of RPE cells precedes the degeneration of photoreceptors. This may reflect an opportunity for therapeutic intervention window directed towards the RPE cells. Our current study supports the potential of the use of RPE cells as “disease-in-a-dish” biological system to test emerging therapies for *ABCA4*-mediated retinopathies. 

## 5. Conclusions

The RPE is thought to protect the retina from inappropriate attacks of the innate immune reactivity by expressing complement negative regulatory proteins. While genetic association studies have established the role of complement system in the etiology of maculopathies, such role for the RPE has never been shown in STGD1. Here, we recapitulated key pathological features of STGD1 in cultured RPE cells derived from a STGD1 patient. STGD1 RPE cells showed lipofuscin-mediated complement activation and improper complement regulation. Decreased inhibition of the complement cascade by CD46 promoted the formation of the membrane attack complex on STGD1 cells causing loss of RPE homeostasis and later cell death. Therefore, mitigation of RPE local complement dynamics may slow disease progression in ABCA4-mediated retinopathies.

## Figures and Tables

**Figure 1 cells-11-03462-f001:**
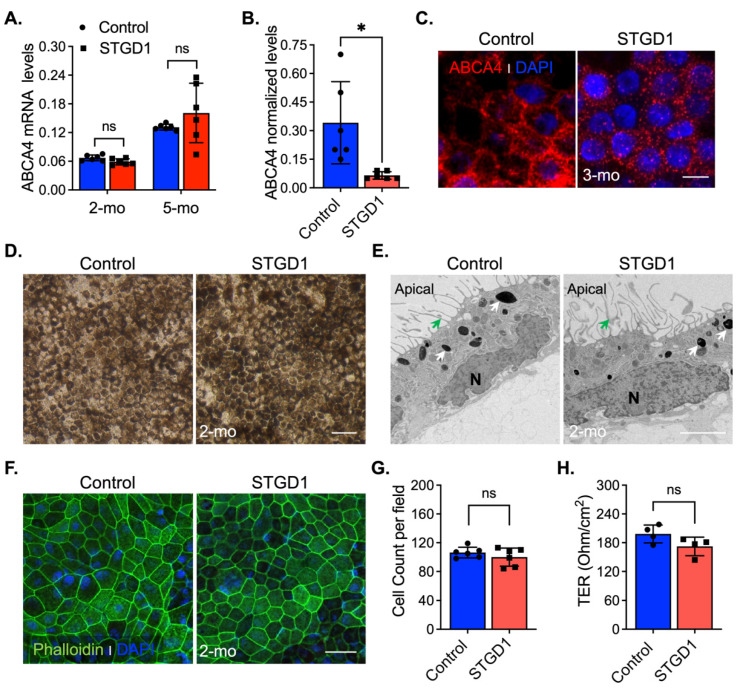
Molecular and morphological features of Control and STGD1 RPE cells. (**A**) Relative levels of ABCA4 mRNA by qRT-PCR from RPE cells for two (2)- and five (5)-months (mo) in culture. (**B**) Normalized ABCA4 protein levels of three-months control and STGD1 RPE cells homogenates (20 μg) by automated WES system (*p* = 0.025). STGD1 RPE cells have 85% reduction in ABCA4 levels compared to control. (**C**) Representative confocal microscopy images of three-months RPE cells fixed and stained for ABCA4 (red, custom-made antibody, generous gift from Dr. Hui Sun). DAPI (blue) stains the nuclei. Note a punctate distribution of ABCA4 throughout RPE cells of STGD1 compared to the control cells. Scale bar = 20 μm. (**D**–**F**) Representative images of two-months of control and STGD1 RPE cells by light (**D**), electron (**E**), and confocal (**F**) microscopy analysis. (**D**) RPE cells of both genotypes display similar cobblestone appearance and homogeneous pigmentation; Scale bar = 100 μm. (**E**) Control and STGD1 RPE cells developed as a monolayer and were polarized presenting apical microvilli (green arrow), melanosomes (white arrow), and nucleus (N). Scale bar = 2 μm. (**F**) Phalloidin staining F-actin filaments (green) evidences a well-defined hexagonal cell shape for both STGD1 vs control RPE cells. DAPI (blue) stains the nuclei. Scale bar = 20 μm. (**G**) Bar graph shows similar number of cells per view field (140 μm^2^) between the control and STGD1 RPE cells (two-months in culture). (**H**) Histogram shows the transepithelial resistance (TER) measured across monolayers of control and STGD1 RPE cells (three-months in culture). Note: All RPE cells were grown in culture with conditioned medium *without* retinoids. In (**A**,**B**,**G**,**H**) data presented as mean ± SD; Statistics analysis using Student’s *t*-test; ns = not significant; * *p* < 0.05; n = 4–6 filter/transwell per genotype.

**Figure 2 cells-11-03462-f002:**
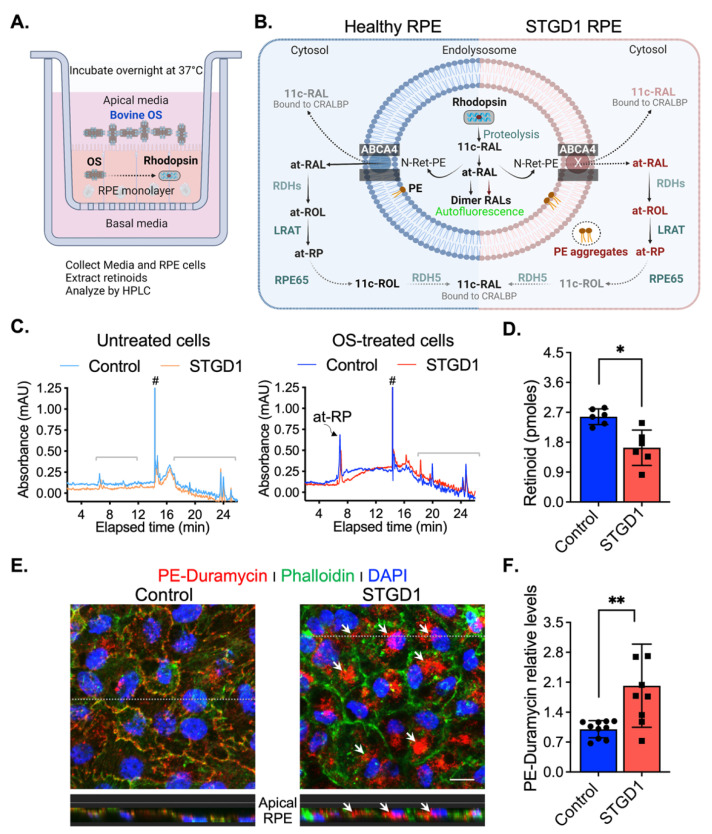
ABCA4-flippase activity is reduced in STGD1 RPE cells. (**A**) Diagram of a transwell with RPE cells grown on a filter to summarize our experimental setting to evaluate recycling of retinaldehyde following overnight incubation with bovine OS containing rhodopsin (retinaldehyde bound with opsin); retinoids were extracted in hexane from media and cell homogenate and analyzed by high-performance liquid chromatography (HPLC). (**B**) Diagram of proposed ABCA4-flippase activity in the endolysosomes of the RPE cell. Following rhodopsin proteolysis, 11-*cis*-retinaldehyde (11c-RAL) and its thermoisomerized form all-trans-RAL (at-RAL) condense with phosphatidylethanolamine (PE) in the endolysosomal membrane to form N-retinyledene-PE (N-Ret-PE). ABCA4 transporter translocates both N-Ret-PE and PE to the cytoplasmic side for recycling. Upon hydrolysis of N-Ret-PE, the 11c-RAL (bound to CRALBP) is readily available to reconstitute the visual pigment, while at-RAL must be reduced by RPE retinal dehydrogenases (RDHs) to all-*trans*-retinol (at-ROL) which is the substrate for lecithin-retinol-acyltransferase (LRAT), the major retinyl ester synthase in RPE cells. LRAT esterifies the at-ROL to all-*trans*-retinyl-palmitate (at-RP), an insoluble storage form of vitamin A, that can be stored or further used as the substrate of RPE65-isomerase to form 11-cis-retinol (11c-ROL) followed by RDH5-oxidation to 11c-RAL. (**C**) Representative HPLC chromatograms at 325 nm of hexane extracts of untreated (left) and OS-treated (right) of three-months control (cyan/blue traces) and STGD1 (orange/red traces) RPE cells. Note that untreated cells (left) have *no* detectable retinoids and all-*trans*-retinyl palmitate (at-RP) is the only detected retinoid in the OS-treated cells (right). Brackets indicate non-retinoid peaks based on spectral analysis and (#) indicates change in solvent gradient. (**D**) Levels of at-RP in the STGD1 RPE cell homogenate were significantly lower vs control. Average data is presented as mean ± SD; Statistics analysis using Student’s *t*-test; * *p* < 0.05 (*p* = 0.0106); n = 6 filter/transwell per genotype. (**E**) Representative confocal merge images of ~six-months control (left) and STGD1 (right) RPE cells that were treated with 1 µM Duramycin (red), specifically labeling the head-group of phosphatidylethanolamines (PE). Phalloidin (green) staining F-actin filaments and nuclei stained by DAPI (blue). Corresponding z-orthogonal representative confocal images are shown below the en-face images. Intracellular PE-aggregates (white arrows) are noticeable in both en-face and z-orthogonal images of STGD1 RPE cells versus control which display significant PE-Duramycin/Phalloidin co-localization (orange-like color area delineating cell boundaries). (**F**) Bar graph shows the relative PE-duramycin levels determined by measuring pixel intensity from acquired confocal images (n = 10 for Control and n = 9 for STGD1). Average data presented as mean ± SD; ** *p* < 0.005 (*p* = 0.0048); n = 3 filter/transwell per genotype; Scale bar = 10 µm. Note: RPE cells were grown in culture for three-months (**C**,**D**) *without* any retinoid supplementation and for six-months (**E**,**F**) with bovine retinal extract supplementation (~1.5 pmoles per feeding, twice per week). Drawings in (**A**) and (**B**) were created with BioRender.com.

**Figure 3 cells-11-03462-f003:**
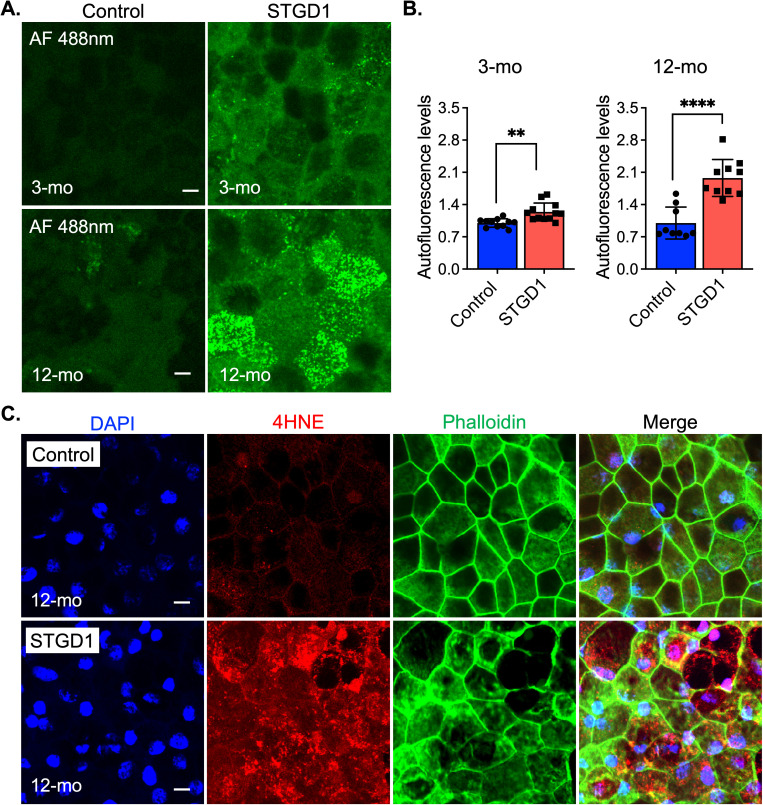
Age-dependent autofluorescent buildup and lipid peroxidation are amplified in STGD1 RPE cells. (**A**) Representative confocal images of autofluorescence (green) acquired with an excitation of 488nm (emission of 515nm) of control (left) and STGD1 (right) RPE cells in culture for three-months (3-mo, top row) and 12-months (12-mo, bottom row) in the presence of bovine retinal extract supplemented at two-months (~1.5 pmoles per feeding). Scale bar = 10 μm. (**B**) Bar graphs show the relative autofluorescence levels determined by measuring pixel intensity from images acquired at 3-mo (left) and 12-mo (right). Statistics analysis using Student’s *t*-test; ** *p* < 0.005 (*p* = 0.0012) and **** *p* < 0.0001; n = 3 filter/transwell per genotype. (**C**) Representative confocal images of 12-months RPE cells fixed and stained with an antibody against 4-hydroxynonenal (4-HNE, red), a natural lipid peroxisomal marker. Phalloidin (green) stains the apical F-actin filaments and DAPI (blue) marks the cell nuclei. Note the intense immunoreactivity for the 4-HNE antibody on the STGD1 RPE cells. Scale bar = 10 μm; n = 3 filter/transwell per genotype.

**Figure 4 cells-11-03462-f004:**
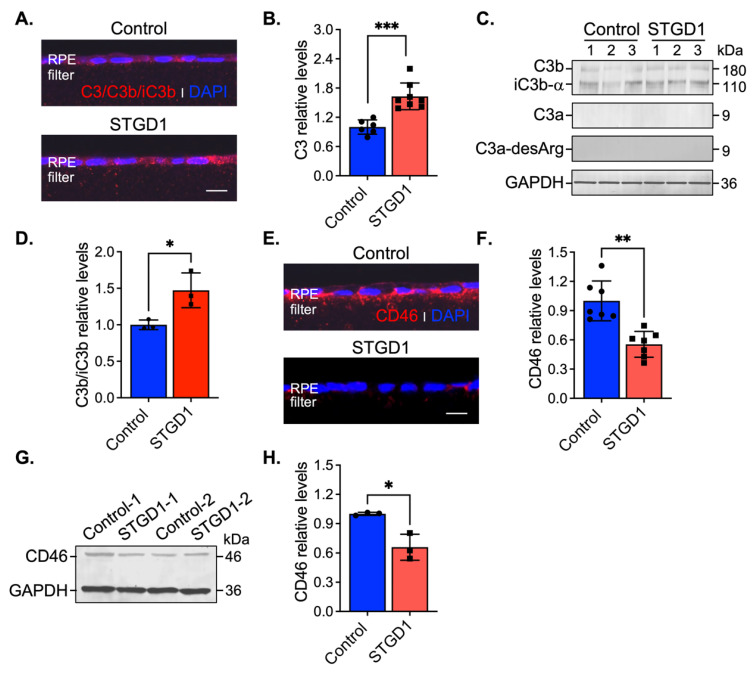
C3 complement dysregulation is observed in STGD1 RPE cells. (**A**) Representative confocal images of C3/C3b/iC3b (red) staining on sections of fixed and agarose embedded RPE cells of control (top) and STGD1 (bottom). (**B**) Histogram shows the relative levels of C3 based on pixel intensity acquired by confocal microscopy. Average data presented as mean ± SD; *** *p* < 0.0005 (*p* = 0.0003); n = 3 filter/transwell per genotype. (**C**) Representative immunoblots of equal amount of RPE cell homogenates of control and STGD1 using C3, C3a-desArg, and GAPDH antibodies. Under reducing conditions, C3 antibody recognizes C3b, iC3b, and C3a fragments. C3a fragment was not detected with either of the C3 or C3a-desArg antibodies. Numbers one to three on each blot represent independent biological samples for each genotype. (**D**) Histogram presents the relative levels of cumulative C3b and iC3b major fragments after normalizing to GAPDH loading control. Average data is presented as mean ± SD; * *p* < 0.05 (*p* = 0.041); n = 3 filter/transwell per genotype. (**E**) Representative confocal images of CD46 (red) staining on sections of RPE cells of control (top) and STGD1 (bottom). Noticeable diminished CD46 staining was observed in STGD1 RPE cells. (**F**) Histogram shows the relative levels of CD46 protein based on pixel intensity acquired by confocal microscopy. Average data is presented as mean ± SD; ** *p* < 0.005 (*p* = 0.0013); n = 3 filter/transwell per genotype. (**G**) Representative immunoblot of 10 μg total protein of RPE homogenates from two biological samples (1- and 2-) consisting of pooled two transwells of RPE cells of control and STGD1, respectively. (**H**) Histogram presents the relative levels of CD46 after normalizing to GAPDH loading control. Average data is presented as mean ± SD; * *p* < 0.05 (*p* = 0.037); n = 3 replicates per genotype. Note: All RPE cells were grown in culture for three-months in the presence of bovine retinal extract supplemented at two-months (~1.5 pmoles per feeding) at regular feeding regimen (twice a week). Immunohistochemistry staining and quantification of C3/C3b/iC3b (**A**,**B**) and CD46 (**E**,**F**) were done using sections from the same transwell (n = 3 filter/transwell per genotype). Experiment was repeated twice. In (**A**,**E**) nuclei were stained with DAPI (blue) and scale bar = 20 μm.

**Figure 5 cells-11-03462-f005:**
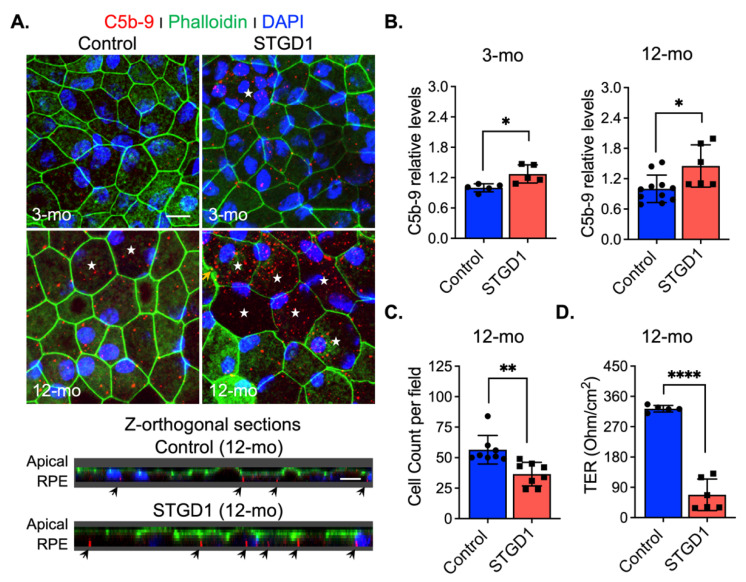
MAC (C5b-9) deposition induces cell damage in STGD1 RPE cells. (**A**) Representative confocal merge images of C5b-9 (red) and Phalloidin (green) acquired of control (left) and STGD1 (right) RPE cells in culture for three-months (3-mo, top row) and 12-months (12-mo, bottom row) in the presence of bovine retinal extract supplemented at two-months (twice per week). Corresponding z-orthogonal sections for 12-mo group are shown below the en-face confocal images. En-face images of 12-mo STGD1 RPE cells displayed numerous larger cells (white stars) delineated by the Phalloidin staining and co-localization of C5b-9 with Phalloidin is indicated by yellow arrows. Abundant C5b-9 immunoreactivity, with predominant basolateral distribution (indicated by the black arrows), is also evidenced in the z-orthogonal sections of 12-mo STGD1 RPE cells. Nuclei were stained with DAPI (blue). Scale bar = 10 μm. (**B**) Bar graphs show the relative C5b-9 levels determined by measuring pixel intensity from images acquired at 3-mo (left) and 12-mo (right). Average data presented as mean ± SD; * *p* < 0.05 (*p* = 0.037 for 3-mo and *p* = 0.015 for 12-mo); n = 3 filter/transwell per genotype. (**C**) Histogram shows average data presented as mean ± SD for total cell count of 12-mo control and STGD1 RPE cells; ** *p* < 0.005 (*p* = 0.001); n = 3 filter/transwell per genotype. (**D**) Transepithelial resistance measurements of 12-mo control and STGD1 RPE cells. Average data presented as mean ± SD; **** *p* < 0.0001; n = 5 filter/transwell for control and n = 6 filter/transwell for STGD1. Statistics analysis using Student’s *t*-test.

**Figure 6 cells-11-03462-f006:**
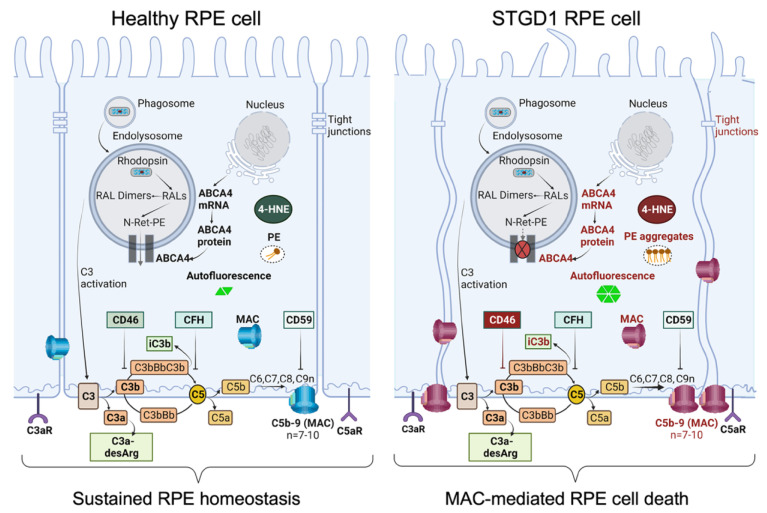
Schematic diagram summarizing our data of healthy RPE (**left**) and STGD1 RPE (**right**). ABCA4-deficiency in STGD1 RPE cells results in the ongoing intracellular accumulation of autofluorescent-lipofuscin material, including phosphatidylethanolamine (PE) aggregates, and augmented C3 complement cascade on the basolateral side of plasma membrane of RPE cells. An early key pathological feature of STGD1 RPE cells is the improper inhibition of C3 convertase by the CD46 complement negative regulator protein. Consequently, CD46 insufficiency in STGD1 RPE cells causes abnormal formation of complement terminal complex (C5b-9) on RPE plasma membrane. With more C9 addition to the C5b-9, the terminal complex destabilizes the plasma membrane by forming a lytic pore which causes loss of the RPE cells in the STGD1 patients. Diagram created with BioRender.com.

## Data Availability

The data presented in this study are available on request from R.A.R. (radu@jsei.ucla.edu).

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
