# Peer review of "Membrane Attack Complex Mediates Retinal Pigment Epithelium Cell Death in Stargardt Macular Degeneration"

_cells, 2022, doi:10.3390/cells11213462_

Round 1

Reviewer 1 Report

This manuscript is an innovative and thorough tour de force which exhaustively compares the structural and biochemical phenotype of iPSC derived RPE cells from a patient with Stargardt disease with two ABCA4 mutations with an unaffected control. This study follows up important discoveries by the senior author that ABCA4 is expressed by the RPE, and this manuscript examines RPE-autonomous pathology in Stargardt. It includes such in depth analysis as one year of functional followup of the control and ABCA4 cells using creative and ingenious quantifiable assays. The data are very robust and include morphometric features of the RPE in these human cells as well as in Abca4-/- mice, functional analysis of ABCA4 activity, loss of a key complement regulatory protein (CD46) and increased membrane attack deposition on the RPE. Also, brilliantly, the Stargardt cells and control cells are essentially identical at the major AMD-associated variants.

The MS is outstanding overall.

My one suggestion is the line in the discussion stating “RPE dysfunction is thought to be the initiator of photoreceptor degeneration in STGD1 and AMD-associated with some ABCA4 610 variants [3].” I think this may depend on the specific mutations in Stargardt, and it is not uncommon to see loss of photoreceptor cells on top of a normal (intact) layer of RPE. This may be related to the kinetics of OS turnover, the degree of ABCA4 loss of function/rate of bisretinoid removal, therefore relating to whether harmful accumulation can take place, and thus relate to whether a patient has Stargardt disease cone rod dystrophy, or pan retinal photoreceptor degeneration/non-macula sparing RP (e.g., Sheffield and Stone NEJM 2011).  The senior author and team are complimented for their huge contribution of our understanding of RPE-expressed ABCA4 and its contribution to disease, but I’d probably change this wording a bit (if the authors agree).

Author Response

We thank Reviewer #1 for the positive remarks of our work, acknowledging the significance of our study, and emphasizing on the multi-dimensional approach (structural, biochemical, and functional) to assess these human-derived RPE cells. In fact, this is the first study describing the RPE only ABCA4-mediated pathology in a newly developed STGD1 RPE “disease-in-a-dish” experimental model.

We appreciate reviewer’s comments regarding the line in the Discussion section and recognize that the wording was unclear. We did not intent to imply that all AMD-related photoreceptors degeneration was due to RPE dysfunction, we only intended to state this is the case when the patient with AMD has very specific ABCA4 variants.

We revised the section to reflect reviewer’s insights: "The initiator of photoreceptor degeneration in STGD1, and in some cases of AMD that are associated with specific ABCA4 variants, is thought to be due to RPE dysfunction." We are very familiar with the Sheffield and Stone paper, and other causes for photoreceptors degeneration in AMD, however, as these studies are centered on Stargardt Disease and ABCA4 pathogenic variants, we wish to keep the discussion limited to this topic. Consequently, we did not include other mechanisms of photoreceptor degeneration in AMD to this section as it is a slightly different topic.

Reviewer 2 Report

Membrane Attack Complex Mediates Retinal Pigment Epithelium Cell Death in Stargardt Macular Degeneration

The manuscript describes an in-vitro RPE cell model of Stargardt disease. The authors are to be complemented on the novel use of iPSC derived RPE cells from Stargardt patients which were supplemented with bovine retinal extracts and all-trans retinol in culture. This set-up was applied to gain new insights into complement-mediated RPE pathology which recapitulates observations in an ABCA4 K/O mouse model. 

Overall, the findings are elegantly presented and provides a step-change in modelling this retinopathy under culture conditions. This is very nice work indeed. The major findings are fairly represented and discussed, though on occasion, there are some over interpretations of results. Specific comments for the authors are as follows. Addressing these would complete what would otherwise be an important paper in the field.    

Page 2, line 85: please indicate the pore size of PET transwell membranes.

Page 4, section 2.9: phagocytosis assay.

It is unclear why a proportion POS remains on the apical RPE surface without internalisation. All POS are expected to bind to MerTK and a5bV integrin and be phagocytosed for degradation. The authors should explain why a synchronised pulse was not used following the method described by Hall and Abrams (PMID: 2828096).    

Fig S1A-B: n=6 transwells/genotype. Is this one experiment? This should be clarified in the legend next to where information related to this panels are described. Details of this type should be included in all the figure legends in the supplement. 

Fig S1E: please consider using higher magnification images to highlight ABAC4 co-localisation with EEAI vesicles.

Page 7, line 311-312: “Together, these findings suggest degradation and mislocalization of the mutated ABCA4 protein”. There is no direct evidence for POS degradation, only co-localisation. How can you say this? 

Figure 1: please provide higher magnification images for panels C-E. Panel B: Fold reduction should also be stated in the figure legend. Panel H: please provide details of n number (experimental replicates and biological repeats).

Page 9, line 357: Why was an early time point such as 4 hours chosen to assess POS degradation? Evidence suggests this takes much longer once RPE cultures are fed POS (as shown by the Finnemann lab).

Figure 2A-B: Please provide sharper/higher resolution images. These are blurry and rather pixilated (unfortunately, some authors use BioRender, which only generates low-resolution images that are wholly unsuited for producing publication-quality images). 

Figure S4A and D: what does 1, 2 and 3 mean in control and STGD1 western blots? Are these different clones? S4G: poor resolution schematic (see comments above).

Figure S5A-B: It is not necessary to highlight staining artefacts in the panel and figure legend, which is distracting from what is otherwise a clear result. Rather, it would be more useful if the take ‘home message’ of this finding was stated in the legend.

S5C: Normalised to what (as stated in the legend)?

Figure S6A: the legend states that bovine retinal extract was supplemented at 2 months. What does that mean? Please clarify for how long this was added to the culture media. Also indicate the supplier of bovine retinal extract in the methods section. The authors go on to state that this was added at ~1.5 pMol per feeding. What does this mean? How frequently was this added to the conditioned media (presumably on transwell inserts)? How frequently was the conditioned media changed? Important experimental details of this kind are missing from this manuscript (see also earlier comments related to missing experimental details of this kind).  

Page 14, line 548: please define what is meant by the lateral poles of cells. Is this referring to apical and basolateral surfaces? 

Figure S6B: orthogonal images should be better annotated to highlight exclusively apical peropsin distribution.

Figure 5A-B: orthogonal sections indicate that C5b-9 is intracellular. Is that correct? What does this suggest to the overall mechanism of complement-mediated damage?   

STGD1 cells are suggested to have thinner cell boundaries (line 564 page 15). Unclear how this can be determine from light microscopy resolution images. Please modify this statement or provide EM data to back-up this claim.  

Figure S7A-B: what does 1, 2 and 3 in WT and K/O mice represent? Are these eyes from different animals? The issue here is similar to what was raised before- insufficient experimental details in the methods section.

Figure S7E: legend should state ‘apparent’ breaks between RPE cells, as this cannot be stated for certain without EM data. 

Figure S7F: please provide better annotated orthogonal images.

Figure S7G: most mouse RPE are binuclear (PMID: 25923208), so please clarify what is meant by multinucleation and changes in these types of cells in this in-vitro RPE model. Also indicate where RPE flatmounts are obtained from (central or peripheral mouse retina).

Page 16, line 606: “……MAC-mediated lysis of cellular plasma membrane with the merging of adjacent RPE cells”.

This is simply overstated. MAC-mediated RPE lysis must be demonstrated, if this statement was to hold any water. Furthermore, potential cell lysis by this mechanism does not necessarily mean adjacent RPE cells merge without further proof. Please provide additional experiment evidence or moderate these statements.   

Page 18, line 712: please remove the word ‘all’, which is superfluous.  

END

Author Response

Please see below the point-by-point Author Response for Reviewer #2 concerns and attached document for supporting figures.

“The manuscript describes an in-vitro RPE cell model of Stargardt disease. The authors are to be complemented on the novel use of iPSC derived RPE cells from Stargardt patients which were supplemented with bovine retinal extracts and all-trans retinol in culture. This set-up was applied to gain new insights into complement-mediated RPE pathology which recapitulates observations in an ABCA4 K/O mouse model. 

Overall, the findings are elegantly presented and provides a step-change in modelling this retinopathy under culture conditions. This is very nice work indeed. The major findings are fairly represented and discussed, though on occasion, there are some over interpretations of results. Specific comments for the authors are as follows. Addressing these would complete what would otherwise be an important paper in the field.”     

Authors Response: We would like to thank Reviewer #2 for the effort dedicated to evaluating our study and acknowledge its importance in the STGD1 field. We greatly appreciate the reviewer’s comments which gave us the opportunity to improve the presentation of our data and clarify some statements. Briefly, we have changed the resolution for some images acquired by confocal and electron microscopy, performed new experiments to further assess ABCA4 colocalization with endolysosomal proteins, revised the text per reviewer’s recommendation, and provided point-by-point responses below:

“Page 2, line 85: please indicate the pore size of PET transwell membranes.”

Authors Response: Pore size of the transwell filter insert is 0.4 mm as indicated in the Material & Methods 2.1 section.

“Page 4, section 2.9: phagocytosis assay. 

It is unclear why a proportion POS remains on the apical RPE surface without internalisation. All POS are expected to bind to MerTK and a5bV integrin and be phagocytosed for degradation. The authors should explain why a synchronised pulse was not used following the method described by Hall and Abrams (PMID: 2828096).”    

Authors Response: We agree with the reviewer that some residual POS were still attached to the RPE cells even at the end of the chase-phase, when cells were no longer exposed to POS. However, this non-specific Rho-bound signal was significantly lower and importantly, it was no different between control and STGD1 cells (Figure S2B). 

We thank the reviewer for mentioning the phagocytosis method described in Hall and Abrams paper. Our protocol is similar to their pulse-chase assay performed at 370C, using rat RPE cells, and updated according to Hazim et al (2017) to optimize for human iPSC-derived RPE cells. Our pulse-chase assay reflected no difference of bound and ingested POS, and importantly, the Rho degradation rate was similar for Control versus STGD1 RPE cells without any pathological features at two-months in culture.

“Fig S1A-B: n=6 transwells/genotype. Is this one experiment? This should be clarified in the legend next to where information related to this panels are described. Details of this type should be included in all the figure legends in the supplement.”  

Authors Response: For each qRT-PCR experiment at 3-mo and 5-mo respectively, we used 6 transwells/genotype. All figure legends have been updated to clarify the number of experiments and n’s.

“Fig S1E: please consider using higher magnification images to highlight ABAC4 co-localisation with EEAI vesicles.” 

Authors Response: During this revision period, we did an experiment using the Deltavision OMX super-resolution microscope for a higher-resolution image analysis using early endolysosomal marker EEA1 and ABCA4. We obtained similar results as previously shown by confocal microscopy analysis of ABCA4/EEA1 co-localizing experiment shown in Figure S1E. However, higher background on the OMX microscope was noted due to scattering of light by the filter that iPSC-derived RPE cells were grown on and due to pigmentation of the RPE cells. ABCA4/RAB5 co-localization was evaluated by confocal microscopy in the Control RPE cells only. Merge-images of green/red channels reflects co-localization of ABCA4/RAB5 similarly to ABCA4/EEA1 in the control RPE cells shown in Figure S1E. RAB5, another early endosomal marker, was previously shown to colocalize with ABCA4 using mouse eye tissue (Lenis et al, 2018).As both experiments involved qualitative analysis, we do not think it adds to our initial report but would like to share the data with the reviewers [see attached  Figure R1 panel A (OMX microscopy - ABCA4/EEA1) and B (confocal microscopy – ABCA4/RAB5)].

“Page 7, line 311-312: “Together, these findings suggest degradation and mislocalization of the mutated ABCA4 protein”. There is no direct evidence for POS degradation, only co-localisation. How can you say this?”  

Authors Response: We would like to clarify the statement that the reviewer is referring on page 7. While ABCA4 mRNA levels seem to be similar in Control and STGD1 cells (Figure 1A), we observed lower ABCA4 levels (Figure 1B) and punctate appearance ABCA4 immunostaining (Figure 1C) in the STGD1 RPE cells suggesting decrease abundance due to protein degradation and possible mislocalization. The degradation in this case refers to ABCA4 protein and not POS-Rho degradation as the RPE cells were characterized in the absence of retinoids and/or POS supplementation. Figure 1 legend includes a note to indicate that “All RPE cells were grown in culture with conditioned medium without retinoids”.

“Figure 1: please provide higher magnification images for panels C-E. Panel B: Fold reduction should also be stated in the figure legend. Panel H: please provide details of n number (experimental replicates and biological repeats).” 

Authors Response: Images by light microscopy (Figure 1D) were replaced with ones at a higher magnification. Confocal images in Figure 1 panel C and F, showing the ABCA4 staining and RPE cells distribution respectively, were provided at the lower magnification to allow evaluation of a larger area of the RPE monolayer. 

EM images shown in Figure 1E were taken at 2,000x magnification to incorporate key morphological features including the nucleus of one RPE cell. While 6,000x higher magnification images were taken by EM, it was difficult to have one RPE cell in frame in a proper orientation. Representative EM images at 6,000x are included now in Supplementary material Figure S1F, and also here for reviewer’s evaluation (see attached  Figure R2).

 We added the following in Figure 1 legend: “In (A), (B), (G) and (H) data presented as mean ± SD; Statistics analysis using Student’s t test; ns = not significant; *p<0.05; n=4-6 filter/transwell per genotype.”

 Also, we updated the legend for Figure 1 panel B to include the fold reduction in protein levels. 

“Page 9, line 357: Why was an early time point such as 4 hours chosen to assess POS degradation? Evidence suggests this takes much longer once RPE cultures are fed POS (as shown by the Finnemann lab).”  

Authors Response: Our protocol was adapted from Hazim et al (2017) and optimized for human iPSC-derived RPE cells. Hall & Abrams (1987) indicated a significant decrease in Rho-degradation following 3 hrs of ingestion. Rho-degradation was also attempted at six-hours chase-phase but Rho-staining was very weak to quantify therefore, we presented the 4-hrs Rho-staining quantification.

“Figure 2A-B: Please provide sharper/higher resolution images. These are blurry and rather pixilated (unfortunately, some authors use BioRender, which only generates low-resolution images that are wholly unsuited for producing publication-quality images).”  

Authors Response: We apologize for the poor quality of the diagrams that must have happened at the journal portal. All our figures were submitted at the resolution required by the publisher. We also uploaded individual figures as Tiff files to assist the reviewer’s evaluation.

“Figure S4A and D: what does 1, 2 and 3 mean in control and STGD1 western blots? Are these different clones? S4G: poor resolution schematic (see comments above).”

Authors Response: For Figure S4A and S4D we indicate in the legend that #1-3 correspond to independent biological samples (RPE homogenate from one transwell with 200,000 RPE cells) from the same line.

“Figure S5A-B: It is not necessary to highlight staining artefacts in the panel and figure legend, which is distracting from what is otherwise a clear result. Rather, it would be more useful if the take ‘home message’ of this finding was stated in the legend. 

S5C: Normalised to what (as stated in the legend)?” 

Authors Response: The (*) artefact staining mark was removed from the images and legend of Figure S5A. We also included the following in the Figure S5 legend: “Note: taken together these findings suggest that C3a/C3aR and C5a/C5aR signaling pathways are not active players to STGD1 pathology. Further, support for CD46 complement dysregulation in STGD1 is signified as levels of CD59 and CFH, both negative complement regulators, are similar in control and STGD1 RPE cells”

Histograms in Figure S5 show relative to the control protein levels (Control at 1) after being normalized to internal markers GAPDH and IRBP for RPE cell homogenates (S5C, D, and F) and IRBP (S5H) for media respectively. Figure S5 legend was updated to incorporate this information.

“Figure S6A: the legend states that bovine retinal extract was supplemented at 2 months. What does that mean? Please clarify for how long this was added to the culture media. Also indicate the supplier of bovine retinal extract in the methods section. The authors go on to state that this was added at ~1.5 pMol per feeding. What does this mean? How frequently was this added to the conditioned media (presumably on transwell inserts)? How frequently was the conditioned media changed? Important experimental details of this kind are missing from this manuscript (see also earlier comments related to missing experimental details of this kind).”  

Authors Response: Materials and Methods first section (2.1) was updated to incorporate details regarding the feeding regimen with bovine retinal extract: “RPE cells were grown in culture using optimal media without retinoids for the initial characterization at two-, three-, and five-months. For longitudinal experiments, media was supplemented with bovine retinal extract, as a source of retinoid (~1.5 pmoles per feeding) for the RPE cells. Supplementation was initiated at two-months under our standard feeding regimen (twice a week) and RPE cells were maintained in culture up to 12-months.”

The bovine retinal extract was prepared using freshly collected retinas from bovine eyes according to published methods (Hu and Bok, 2010). Retinoid content was quantified by HPLC using optimized and previously published protocol (Radu et al, 2008). We updated section (2.1) in the revised manuscript.

“Page 14, line 548: please define what is meant by the lateral poles of cells. Is this referring to apical and basolateral surfaces?”  

Authors Response: We updated the text to indicate a basolateral distribution of the C5b-9 (MAC) staining in respect to apically-stained actin filaments.

“Figure S6B: orthogonal images should be better annotated to highlight exclusively apical peropsin distribution.” 

Authors Response: Green-staining Peropsin protein appears towards the apical side of the cells and above the DAPI-stained nuclei. “Apical” side is indicated in the orthogonal images. Additionally, the figure legend was revised to indicate the basolateral distribution of MAC compared to apically stained Peropsin.  

“Figure 5A-B: orthogonal sections indicate that C5b-9 is intracellular. Is that correct? What does ths suggest to the overall mechanism of complement-mediated damage?”   

Authors Response: The reviewer is correct about internalization of C5b-9. This C5b-9 complex internalization has been observed in RPE cells by Georgiannakis et al (2015). As a phagocytic cell, the RPE presumably internalizes the C5b-9 to prevent further growing of C5b-9 by the addition of more C9 molecules to create a lytic pore.

“STGD1 cells are suggested to have thinner cell boundaries (line 564 page 15). Unclear how this can be determine from light microscopy resolution images. Please modify this statement or provide EM data to back-up this claim.”   

Authors Response: We agree with the reviewer that the term “thinner” was not an appropriate to define the cell boundaries in the absence of higher magnification by an EM analysis. We revised the sentence.  

“Figure S7A-B: what does 1, 2 and 3 in WT and K/O mice represent? Are these eyes from different animals? The issue here is similar to what was raised before- insufficient experimental details in the methods section.” 

Authors Response: This information was indeed omitted for Figure S7A-B and we thank the reviewer for pointing out to us. We have now indicated in the Figure S7 legend that #1-3 samples correspond to independent biological samples (RPE homogenates from three different mice for each genotype).

“Figure S7E: legend should state ‘apparent’ breaks between RPE cells, as this cannot be stated for certain without EM data.”  

Authors Response: Text was revised per reviewer’s suggestion. 

“Figure S7F: please provide better annotated orthogonal images.” 

Authors Response: Orthogonal images were annotated to indicate the polarity of the RPE cells. Thanks.

“Figure S7G: most mouse RPE are binuclear (PMID: 25923208), so please clarify what is meant by multinucleation and changes in these types of cells in this in-vitro RPE model. Also indicate where RPE flatmounts are obtained from (central or peripheral mouse retina).” 

Authors Response: The reviewer is correct about mouse binuclear RPE cells with various distribution in the central vs peripheral region. All images of RPE flatmounts were taken from the superior temporal region at equal distance from the optic nerve as indicated in section 2.10 under Materials and Methods. We defined a multinucleate cell as any mouse RPE cell with more than three nuclei, and updated the figure legend accordingly.  

“Page 16, line 606: “……MAC-mediated lysis of cellular plasma membrane with the merging of adjacent RPE cells”.

This is simply overstated. MAC-mediated RPE lysis must be demonstrated, if this statement was to hold any water. Furthermore, potential cell lysis by this mechanism does not necessarily mean adjacent RPE cells merge without further proof. Please provide additional experiment evidence or moderate these statements. “  

Authors Response: Statement was revised to indicate that amplified terminal complement complex may compromise the RPE cellular plasma membrane. 

“Page 18, line 712: please remove the word ‘all’, which is superfluous.”  

Authors Response: Agreed. “All” was removed from the first sentence in the last paragraph in the Discussion section. 
